# Progress or Regress? Self-Improvement Reversal in Post-training

**Ting Wu** [1 2]  **Xuefeng Li** [2 3]  **Pengfei Liu** [2 3 4]

## Abstract

Self-improvement through post-training methods such as iterative preference learning has been acclaimed for enhancing the problem-solving capabilities (e.g., mathematical reasoning) of Large Language Models (LLMs) without human intervention. However, as exploration deepens, it becomes crucial to assess whether these improvements genuinely signify progress in solving more challenging problems or if they could lead to unintended regressions. To address this, we propose a comprehensive evaluative framework that goes beyond the superficial pass@1 metric to scrutinize the underlying enhancements of post-training paradigms for self-improvement. Through rigorous experimentation and analysis across diverse problem-solving tasks, the empirical results point out the phenomenon of *self-improvement reversal*, where models showing improved performance across benchmarks will paradoxically exhibit declines in broader, essential capabilities, like output diversity and out-of-distribution (OOD) generalization. These findings indicate that current self-improvement practices through post-training are inadequate for equipping models to tackle more complex problems. Furthermore, they underscore the necessity of our critical evaluation metrics in discerning the *progress or regress* dichotomy for self-improving LLMs.

## 1. Introduction

In the rapidly evolving landscape of artificial intelligence (AI), the pursuit of self-improving large language models (LLMs) has garnered significant attention (Singh et al., 2023; Huang et al., 2023; Sun et al., 2024). The essence of self-improvement in LLMs lies in their capacity to iteratively refine models' own performance without hu-

man intervention (Zelikman et al., 2022; Yuan et al., 2024). This capability is paramount as it holds the promise of fostering the development of more autonomous, adaptable, and efficient AI systems (Silver et al., 2016). Embracing and implementing self-improvement methodologies enables us to push the boundaries of these models' capabilities, ultimately fostering the creation of more sophisticated and versatile AI applications (Significant-Gravitas, 2023).

Building on the concept of self-training (Grandvalet & Bengio, 2004), wherein models bootstrap their own generated responses for iterative training, a synergetic effect is observed and amplified. When models produce superior responses, the quality of the training data used to refine the models improves, subsequently enabling even better responses in future iterations. Such iterative post-training has become the standard paradigm for current self-improving AI (Yuan et al., 2024). Notably, STaR (Zelikman et al., 2022) has demonstrated that leveraging model's self-generated reasoning steps for iterative supervised fine-tuning (SFT) can effectively enhance its reasoning abilities. Recent studies (Pang et al., 2024) have further revealed that employing iterative preference optimization in LLMs can achieve more performance improvements in reasoning tasks.

Despite the various post-training methods for self-improvement, there remains a lack of understanding regarding their effects and underlying mechanisms. Therefore, in this study, we first endeavor to provide a comprehensive overview of the main iterative post-training paradigms for self-improvement, identifying the factors that contribute to consistent performance improvements. We decouple the influencing factors into the initial model, task datasets, the number of iterations, and the specific post-training methods employed. By isolating these variables, our comprehensive experiments and analysis uncover their individual and combined effects on the model's performance. This provides actionable insights for practitioners on how to perform iterative self-improvement practices more effectively.

While our extensive empirical results show that all these iterative post-training methods can achieve notable improvements in pass@1 accuracy across various problem-solving benchmarks, the evaluation has been limited to this single and superficial metric. Amidst the quest for self-improvement in LLMs, the persistent question arises:

[1]Fudan University [2]Generative AI Research Lab [3]Shanghai Jiao Tong University [4]Shanghai Artificial Intelligence Laboratory. Correspondence to: Ting Wu <tingwu21@m.fudan.edu.cn>.

AI for MATH Workshop at ICML 2024. Copyright 2024 by the author(s).

**are these iterative post-training methods truly fostering progress, or are they inadvertently leading to regression?** Transitioning beyond using pass@1 accuracy as the indicator of improvement, we further develop an evaluative framework equipped with a comprehensive suite of metrics to assess improvement problems, solutions diversity, and OOD capabilities within the iterative process, enabling us to scrutinize the actual improvements beneath self-improvement. Surprisingly, our evaluation results display a paradoxical trend: as pass@1 accuracy increases, the proposed metrics exhibit consistent performance declines.

The perceived reversals in our evaluative framework prompt a critical reflection on the effectiveness of current self-improvement practices. Through this study, we aim to illuminate the path forward for developing truly self-improving LLMs that balance accuracy, diversity, and robustness. To summarize, our work makes three significant contributions as follows:

• **Comprehensive Analysis:** In Sections 3 and 4, we systematically formulate current post-training methodologies and perform extensive experiments to examine how various factors influence self-improvement in solving challenging tasks. To the best of our knowledge, this work is the first to provide an in-depth overview of these influencing factors.

• **Evaluative Innovations:** In Section 5, we propose a holistic set of evaluation metrics to better capture the multi-faceted nature of LLM performances in self-improvement practices.

• **Critical Insights:** In Section 5, based on the evaluation metrics, we identify the phenomenon of *self-improvement reversal*, where increases in pass@1 accuracy comprise other essential capabilities like solution diversity and OOD generalization.

## 2. Background and Related Work

Training paradigms for LLMs typically consist of two stages: pre-training and post-training (Liu et al., 2024). Common post-training methods include supervised fine-tuning (Taori et al., 2023; Wang et al., 2023) and preference learning (Ouyang et al., 2022; Lee et al., 2024). Supervised fine-tuning trains LLMs to produce standard responses for given instructions, while preference learning trains LLMs to align with human preferences for different responses. Both methods, however, rely heavily on extensive human-annotated data.

An important question is whether effective LLM post-training can be achieved without excessive external feedback. Predating the era of LLMs, the self-training algorithm (Grandvalet & Bengio, 2004; Goodfellow et al., 2014) demonstrated the potential to enhance model performance

without additional labeled data. Recent studies have revived this concept, employing iterative self-training to facilitate self-improvement in LLMs without external feedback (Wang et al., 2023; Sun et al., 2023). For instance, STaR (Zelikman et al., 2022) shows that iterative training on the model's own reasoning traces for correct answers can help solve increasingly difficult problems. Unlike the iterative nature of SFT, recent works (Yuan et al., 2024; Pang et al., 2024) propose iterative preference fine-tuning to aid models in self-improving.

In contrast to post-training methods, another line of research explores self-improvement through iterative post-prompting during inference (Huang et al., 2023). This approach does not update the model's parameters but achieves self-improvement by generating reflections on its outputs and adjusting future outputs accordingly (Madaan et al., 2023; Gou et al., 2024). However, as revealed by Huang et al. (2024), post-prompting strategies are limited by the model's *intrinsic self-correction* capabilities, thereby failing to significantly enhance problem-solving capabilities.

The potential of iterative post-training for self-improvement in LLMs remains underexplored. Although various post-training methods have demonstrated promise in general instruction-following tasks (Li et al., 2024; Sun et al., 2024; Chen et al., 2024; Yuan et al., 2024), they predominantly focus on aligning models with human values rather than enhancing the models' internal knowledge. A key challenge remains whether LLMs can sustain consistent performance on more complex problem-solving tasks. Recently, Pang et al. (2024) examined iterative preference learning in the context of reasoning tasks, marking the first study to expand beyond instruction tuning.

Despite these advancements, a comprehensive overview investigating the effectiveness of various iterative post-training methods for problem-solving is still lacking. **First**, it remains unclear how improvements vary across iteration steps, different base models, task difficulties, and iterative post-training techniques. For practitioners, there is a need for guidelines to help choose the most effective post-training method among the various iterative post-training paradigms. **Second**, current research only concentrates on maximizing benchmark scores through iterative self-improvement, there is little exploration of the underlying factors contributing to performance gains. As a result, the progress and reliability of different self-improvement methods are not guaranteed.

In this work, we aim to address these two critical issues. Our goal is not only to ensure the effectiveness of various self-improvement methods but also to ensure that other capabilities are not compromised during the complex self-improvement process.

# 3. Post-training for Self-Improvement

## 3.1. Formulation

Consider a training dataset $\mathcal{D} = \{(x_i, y_i)\}_{i=1}^N$, consisting of pairs of queries $x_i$ and their corresponding correct responses $y_i$. A foundation model, denoted as $M_0$. Our objective is to enhance $M_0$ through a self-driven iterative post-training process, leveraging the model's own outputs to refine its capabilities, without reliance on external signals.

**Iterative Post-Training**   The iterative post-training process involves a series of *post-training steps*, each aimed at using the model's previous outputs to guide its subsequent refinement. These steps are designed to foster a continuous loop of self-improvement for the model.

The process is outlined across three main phases as follows, where the total number of iterations is denoted as $T$, and the model employed in the $t$-th iteration is denoted as $M_{t-1}$, implying that $M_0$ is used in the first iteration

  • **Answer sampling**: In the $t$-th iteration, we prompt $M_{t-1}$ to generate $N$ answers for each query $x_i$ in $\mathcal{D}$ to form a new self-generated dataset $\mathcal{D}_t^{\text{self}} = \{(x_i, y_i^j)|x_i \in \mathcal{D}, j = [1, N]\}$.

  • **Training set construction**: The training set $D_t$ in the $t$-th iteration is assembled from $\mathcal{D}_t^{\text{self}}$ without introducing any external data. The approach to constructing the training set depends on the specific paradigm of post-training.

  • **Model post-training**: Utilizing $\mathcal{D}_t$, the model $M_{t-1}$ is refined into $M_t$.

It's worth noting that, in the first iteration, we always directly supervised fine-tuning $M_0$ on $\mathcal{D}$ to initialize $M_1$ with task-specific knowledge.

Central to these diverse methodologies is the post-training function, symbolized as $\mathcal{F}$. We distinguish among the practices based on the nature of $\mathcal{F}$, involving Supervised Fine-tuning (SFT) and Direct Preference Optimization (DPO), the latter being an effective implementation of preference learning. During the SFT phase, this stage necessitates accurately labeled training data. We derive these correct answers from $\mathcal{D}_t^{\text{self}}$ to assemble the training dataset:

$$\mathcal{D}_t = \{(x_i, y^{\checkmark})|\mathcal{R}(x_i, y^{\checkmark}) = 1, (x_i, y^{\checkmark}) \in \mathcal{D}_t^{\text{self}}\}$$

where $\mathcal{R}(x, y)$ evaluates whether the answer $y$ accurately addresses the question. In our problem-solving task, the correctness of an answer $y$ is verified by its alignment with the response provided in the dataset. While during the DPO phase, for each query $q_i$ in dataset $\mathcal{D}$, both correct and incorrect responses from $\mathcal{D}_t^{\text{self}}$ are paired to construct the training set, allowing for contrastive preference learning:

$$\mathcal{D}_t = \{(x_i, y^{\checkmark}, y^{\times})|\mathcal{R}(x_i, y^{\checkmark}) = 1, \mathcal{R}(x_i, y^{\times}) = 0,$$
$$(x_i, y^{\checkmark}) \in \mathcal{D}_t^{\text{self}}, (x_i, y^{\times}) \in \mathcal{D}_t^{\text{self}}\}$$

## 3.2. Three Iterative Post-Training Paradigms

---

**Algorithm 1** Iterative Self-Improvement

---

1: training set $\mathcal{D} = \{x_i, y_i\}$, base model $M_0$, iteration times $T$, post-training function series $[\mathcal{F}_1(\cdot), \mathcal{F}_2(\cdot), ..., \mathcal{F}_T(\cdot)]$
1: $M_1 \leftarrow \text{SFT}((M_0)|\mathcal{D})$
2: **for** $t = 2$ to $T$ **do**
3:   $\mathcal{D}_t^{\text{self}} = \{(x_i, y_i^j)|x_i \in D, y_i^j \sim M_{t-1}(x_i), j \in [1, N]\}$
4:   **if** $\mathcal{F}(\cdot) == \text{SFT}$ **then**
5:     $\mathcal{D}_t = \{(x_i, y^{\checkmark})|\mathcal{R}(x_i, y^{\checkmark}) = 1, (x_i, y^{\checkmark}) \in \mathcal{D}_t^{\text{self}}\}$
6:   **else**
7:     $\mathcal{D}_t = \{(x_i, y^{\checkmark}, y^{\times})|\mathcal{R}(x_i, y^{\checkmark}) = 1, \mathcal{R}(x_i, y^{\times}) = 0, (x_i, y^{\checkmark}) \in \mathcal{D}_t^{\text{self}}, (x_i, y^{\times}) \in \mathcal{D}_t^{\text{self}}\}$
8:   **end if**
9:   $M_t \leftarrow \mathcal{F}_t(M_{t-1}|\mathcal{D}_t)$
10: **end for**=0

---

Through the implementation of designated self post-training steps (e.g., self-SFT), several distinct iterative post-training paradigms emerge. Our work focuses on three paradigms: (i) iterative SFT, where each cycle consists exclusively of self-SFT steps, (ii) Iterative DPO, characterized by successive self-DPO steps, except for the first iteration which supervised fine-tune the base model $M_0$, and (iii) iterative SFT-DPO, which initiates with a self-SFT step and alternates between self-DPO and self-SFT steps to form a complete iterative post-training loop.

We describe the unified procedure in Algorithm 1.

# 4. Experiment

As outlined in Algorithm 1, we characterize the key variables—foundation model ($M$), task dataset ($\mathcal{D}$), iteration steps ($T$), and post-training method ($\mathcal{F}$)—critically influence model performance during iterative self-improvement. This section explores the impact of these variables on different problem-solving tasks. We aim to uncover the trade-offs and comparative advantages of Iterative SFT, Iterative DPO, and Iterative SFT-DPO in enhancing performance across various tasks. Through this analysis, we seek to provide deeper insights into the mechanisms driving iterative self-improvement.

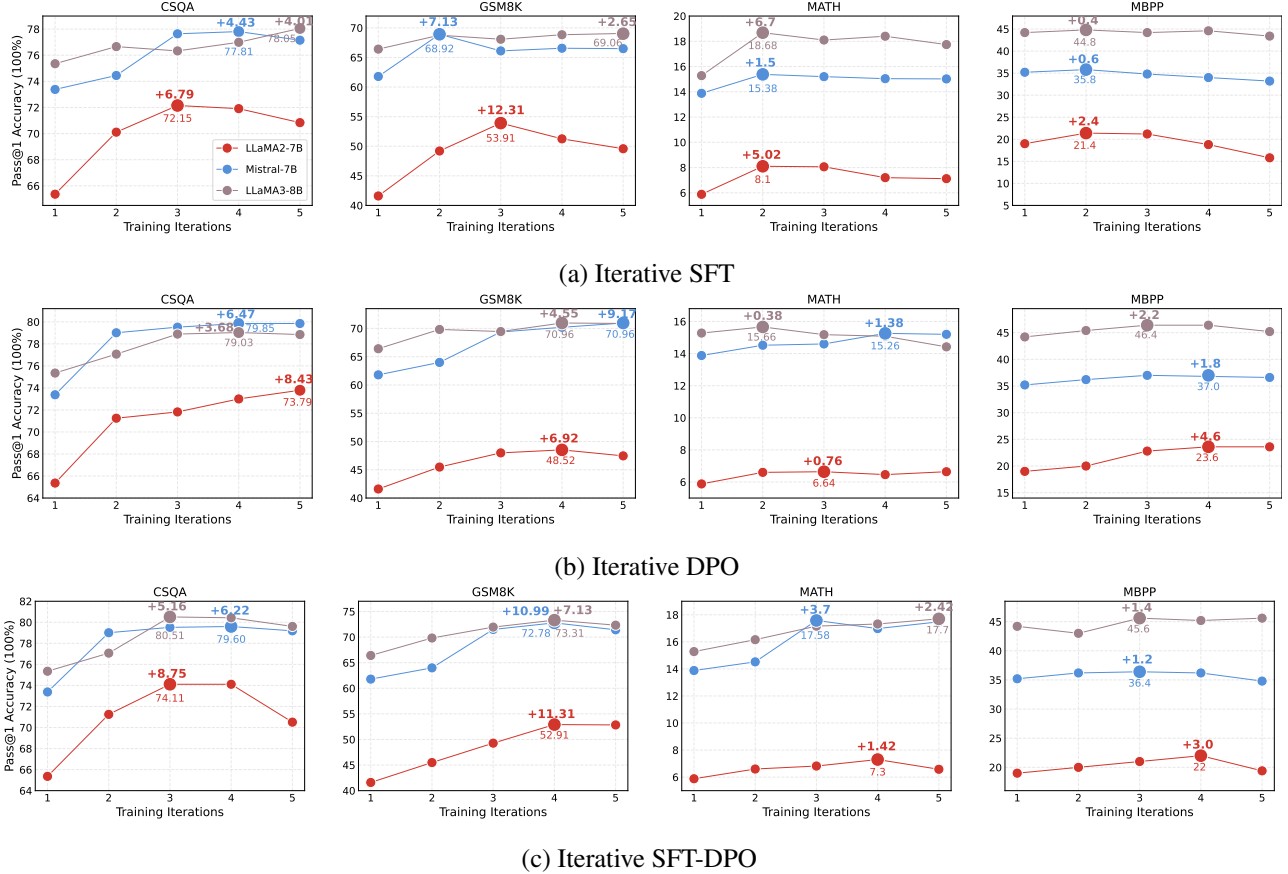

(a) Iterative SFT

(b) Iterative DPO

(c) Iterative SFT-DPO

Figure 1: **Pass@1** accuracy across the four benchmarks by performing with the three paradigms: Iterative SFT, Iterative DPO and Iterative SFT-DPO. For each model along with the training iterations, we highlight the optimal result with a larger-size marker, the improvement above and final accuracy below.

### 4.1. Experimental Setup

**Datasets** To measure model problem-solving capabilities, we train and test on a broad spectrum of problem-solving datasets. We measure general knowledge using the CommonsenseQA (CSQA) dataset (Talmor et al., 2019), assessing mathematical reasoning with the GSM8K (Cobbe et al., 2021) and MATH (Hendrycks et al., 2021) dataset, and weigh code generation skills using the MBPP dataset (Austin et al., 2021). Regarding the train-test split, we adhere to (Kojima et al., 2022), utilizing the validation set of CommonsenseQA for evaluation. The GSM8K and MATH datasets are employed with their predefined train-test splits. For the MBPP code dataset, we follow the approach outlined by (Austin et al., 2021) that utilizes examples of Task IDs 11-510 as the 500 test problems, and the remaining 374 examples ranging Task IDs from 601 to 974 for fine-tuning.

**Sampling and Rewarding** At the end of training, we sample N=50 outputs for each problem using top p sampling (Holtzman et al., 2020) with $p = 0.95$ and temperature $0.75$. Considering the gold labels are provided for the problem-solving datasets, we use the correctness of final answer as a binary reward for the output to annotate the preference.

**Training** Our experiments primarily leverages three open-source models LLaMA-2-7B (Touvron et al., 2023), Mistral-7B (Jiang et al., 2023) and LLaMA3-8B (AI@Meta, 2024), with a fully fine-tuning setting. For the implementation of preference-based learning, we utilize Direct Preference Optimization (DPO (Rafailov et al., 2023)) due to its scalability and efficiency. In each iteration, preference data are derived by sampling outputs from the newly updated model, utilizing an online sampling strategy. Hence, we posit that this online DPO can be treated as an effective and representative implementation for preference learning (Tajwar et al., 2024).

**Evaluation** We use greedy decoding as the temperature set 0 for testing generation. Meanwhile, we utilize zero-shot prompting (Kojima et al., 2022) for both answer sampling and evaluations since we find for LLMs finetuned on specific

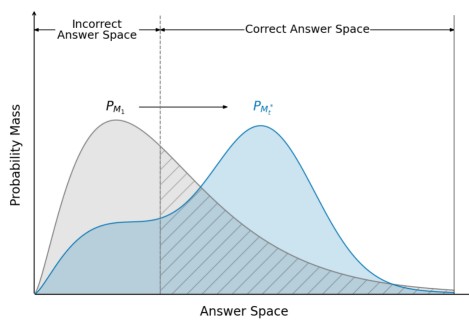

| Model | Dataset | Coverage | $M_1$ | $M_t^*$ Iterative SFT | $M_t^*$ Iterative DPO | $M_t^*$ Iterative SFT-DPO |
|---|---|---|---|---|---|---|
| LLaMA2-7B | CSQA | 0.828 | 65.36 | 72.15 | 73.39 | 74.11 |
| | GSM8K | 0.398 | 41.6 | 53.91 | 48.52 | 52.91 |
| | MATH | 0.167 | 5.88 | 8.1 | 6.64 | 7.3 |
| | MBPP | 0.539 | 19.0 | 21.4 | 23.6 | 22.0 |
| Mistral-7B | CSQA | 0.875 | 73.38 | 77.81 | 79.85 | 79.6 |
| | GSM8K | 0.627 | 61.79 | 68.92 | 70.96 | 72.78 |
| | MATH | 0.143 | 11.56 | 15.38 | 15.26 | 17.58 |
| | MBPP | 0.629 | 35.2 | 35.8 | 37.0 | 36.4 |
| LLaMA3-8B | CSQA | 0.825 | 75.35 | 78.05 | 79.03 | 80.51 |
| | GSM8K | 0.802 | 66.41 | 69.06 | 70.96 | 73.31 |
| | MATH | 0.281 | 15.28 | 18.68 | 15.66 | 17.7 |
| | MBPP | 0.545 | 44.2 | 44.8 | 46.4 | 45.6 |

Figure 2: **Left:** The answer distributions of models. $P_{M_1}$ and $P_{M_t^*}$ represent the answer distributions of $M_1$ and the optimal model $M_t^*$ (achieving the highest pass@1 accuracy) within iterative process. The shaded area indicates the *correct answer coverage* of $M_1$. **Right:** For foundation model $M$ and task $\mathcal{D}$, each line lists the correct answer coverage and the optimal pass@1 accuracy of $M_t^*$ with the three iterative post-training methods. This table aims to display the relationship between correct answer coverage and the effectiveness of the post-training method $\mathcal{F}$.

tasks, zero-shot prompting is superior to few-shot prompting. More experimental details can be seen in Appendix B.

## 4.2. Main Results: Decoupling the Influences of Variables

We perform the three post-training paradigms with the selected LLMs, training and testing them on the respective tasks. Based on the results shown in figure 1, we delve into the detailed analysis of how these variables influence the effectiveness of self-improvement.

**Iteration** $T$ Across all methods and datasets, there is a general trend of improvement in pass@1 accuracy with increasing iteration steps. This indicates that iterative post-training effectively enhances model performance over time. However, the rate of improvement tends to plateau or even decline slightly after 4-5 iterations. This suggests that current post-training methods struggle to achieve long-lasting improvements, and excessive post-training (beyond a certain number of iterations) may even yield diminishing returns.

**Foundation Model** $M$ The optimal accuracy improvements across various datasets and post-training methods suggest that LLaMA2-7B demonstrates a relatively higher capacity for improvement under iterative post-training. For instance, on the GSM8K dataset, LLaMA2-7B with Iterative SFT shows an improvement of +12.31 after 5 iterations, whereas LLaMA3-8B exhibits only a moderate gain. This indicates that the more capable $M_1$ is not necessarily the model that achieves the most significant performance gains during the self-improvement process. However, the most capable model $M_1$ generally achieves the highest optimal accuracy overall. For example, although LLaMA2-7B achieves the maximum gains on GSM8K with Iterative SFT, it still struggles to outperform LLaMA3-8B in terms of absolute

optimal accuracy (53.91 vs. 69.06).

**Problem-solving Tasks** $\mathcal{D}$ Models utilizing the three post-training methods all demonstrate notable improvements on the CSQA and GSM8K datasets, while showing more modest gains on the MATH and MBPP datasets. This indicates that, from the perspective of task difficulty, problems in the CSQA and GSM8K datasets are relatively easier for the models to resolve. In contrast, the MATH dataset poses significant challenges for 7B models due to its complexity. Additionally, the task of code generation, as represented by the MBPP dataset, is also difficult for these foundation models since they were not specifically pretrained on code domains.

**Comparative Analysis of Post-Training Methods** With foundation model $M$ and task $\mathcal{D}$ varying, the best-performing iterative method also changes accordingly. For example, for Mistral-7B on the CSQA dataset, Iterative-DPO achieves the highest accuracy improvement of +6.47. However, when applied to the GSM8K dataset, the Iterative SFT-DPO method yields the maximum improvement of +10.99. Therefore, with these identifiable variables characterized, it remains challenging for downstream practitioners to determine the optimal post-training method $\mathcal{F}$ for their specific use case.

**Answer Coverage: Characterizing More Deciding Factor** As discussed above, the identifiable variables fail to provide clear clues on the effectiveness of the post-training method $\mathcal{F}$ when foundation model $M$ and task $\mathcal{D}$ change. Upon closer examination of Figure 1, we find a common thread: regardless of the changes in $M$ and $\mathcal{D}$, models ($M_1$) that perform well on a task after the initial iteration of SFT tend to show substantial improvements with further iterations by performing iterative DPO and iterative SFT-DPO, compared to using Iterative SFT. Conversely, those $M_1$ that

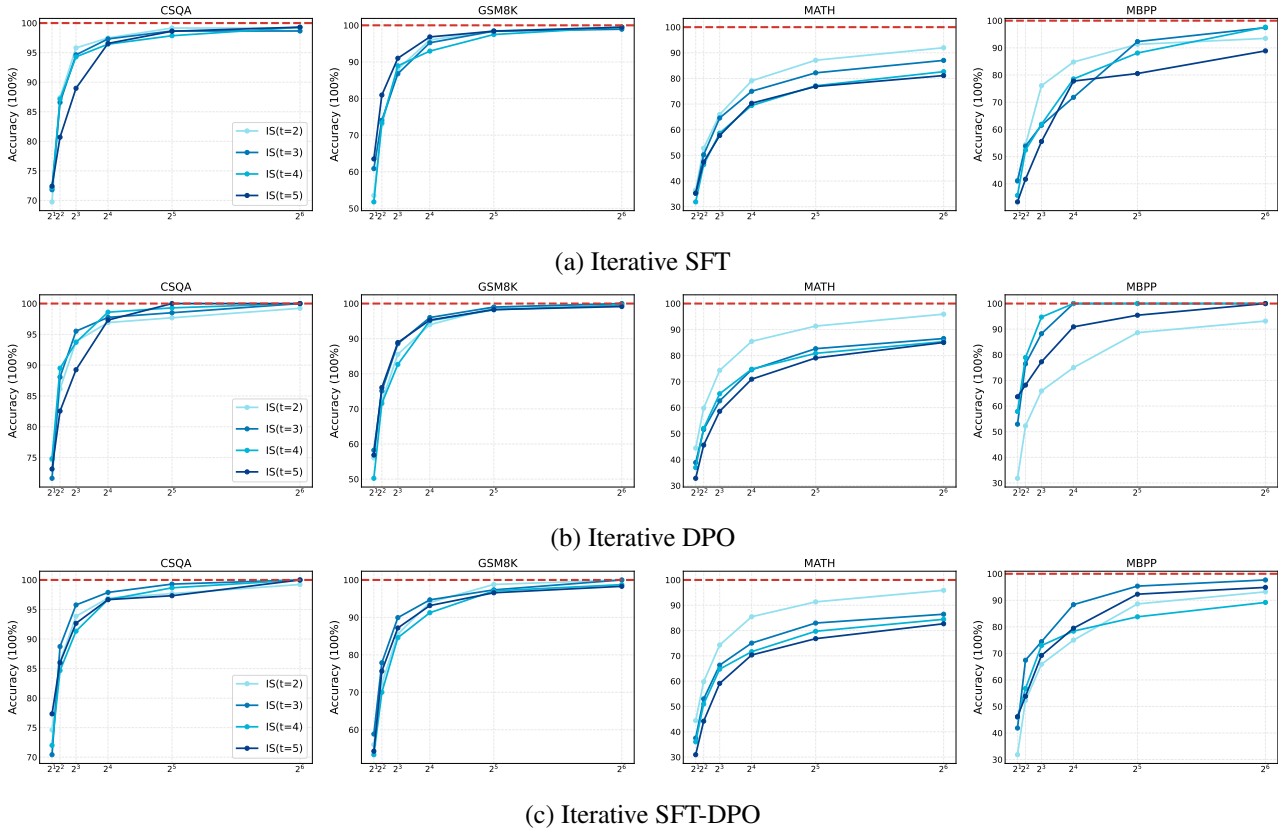

(a) Iterative SFT

(b) Iterative DPO

(c) Iterative SFT-DPO

Figure 3: Pass@N accuracy of $M_1$ with zero-shot prompting on IS(t), for $t > 2$.

achieve lower pass@1 accuracy initially exhibit limited gains with iterative DPO. Based on this observation, we hypothesize that $M_1$'s capability to solve the test problems fundamentally influences further improvement trends and optimal improvements of $\mathcal{F}$. To quantify $M_1$'s capability on the test set, we introduce **Correct Answer Coverage** as a measurement, the proportion of the correct answer space that the model's responses occupy. An illustrative display of this coverage is shown in Figure 2.

Mathematically, we can sample $N$ model's outputs to approximate the answer space. As $N \to \infty$, these outputs can effectively represent the entire answer space. Therefore, expected accuracy over the $N$ outputs can serve as an unbiased estimate of the *correct answer coverage*. Formally, we use the following equation to calculate $M_1$ correct answer coverage (for a more detailed derivation, please refer to the Appendix C.):

$$\text{Correct Answer Coverage} = \mathbb{E}\left[\frac{N_{\text{correct}}}{N}\right]$$

$$\approx \frac{1}{|\mathcal{D}_{\text{test}}|}\sum_{x \in \mathcal{D}_{\text{test}}}\frac{1}{N}\sum_{i=1}^{N}\mathbb{I}[M(x_i) == y] \quad (1)$$

As shown in Figure 2, the relationship between correct

answer coverage and optimal performance of $\mathcal{F}$ validate our prior observation and hypothesis. The table clearly demonstrates that when the correct answer coverage is high ($> 0.5$), Iterative DPO and Iterative SFT-DPO produce the best-performing $M_t^*$. Conversely, when the coverage is lower ($\leq 0.5$), Iterative SFT is more effective in achieving the optimal $M_t^*$. Therefore, correct answer coverage can serve as a key factor in guiding practitioners to choose the most suitable iterative post-training method $\mathcal{F}$ for the specific problem-solving task with a fixed foundation model.

## 5. Critical Evaluations on Self-Improvement

Despite the extensive exploration of various post-training practices for self-improvement and a deepened understanding of their efficacy, current endeavors remain narrowly focused on enhancing performance numbers across these problem-solving benchmarks. Transitioning beyond using pass@1 accuracy as the indicator of improvement, our objective in this section is to engage in a critical examination and reevaluation of iterative self-improvement: discerning whether the improvements constitute genuine progress or merely regression. For brevity, all the results shown in this section is based on the foundation model $M$ as Mistral-7B.

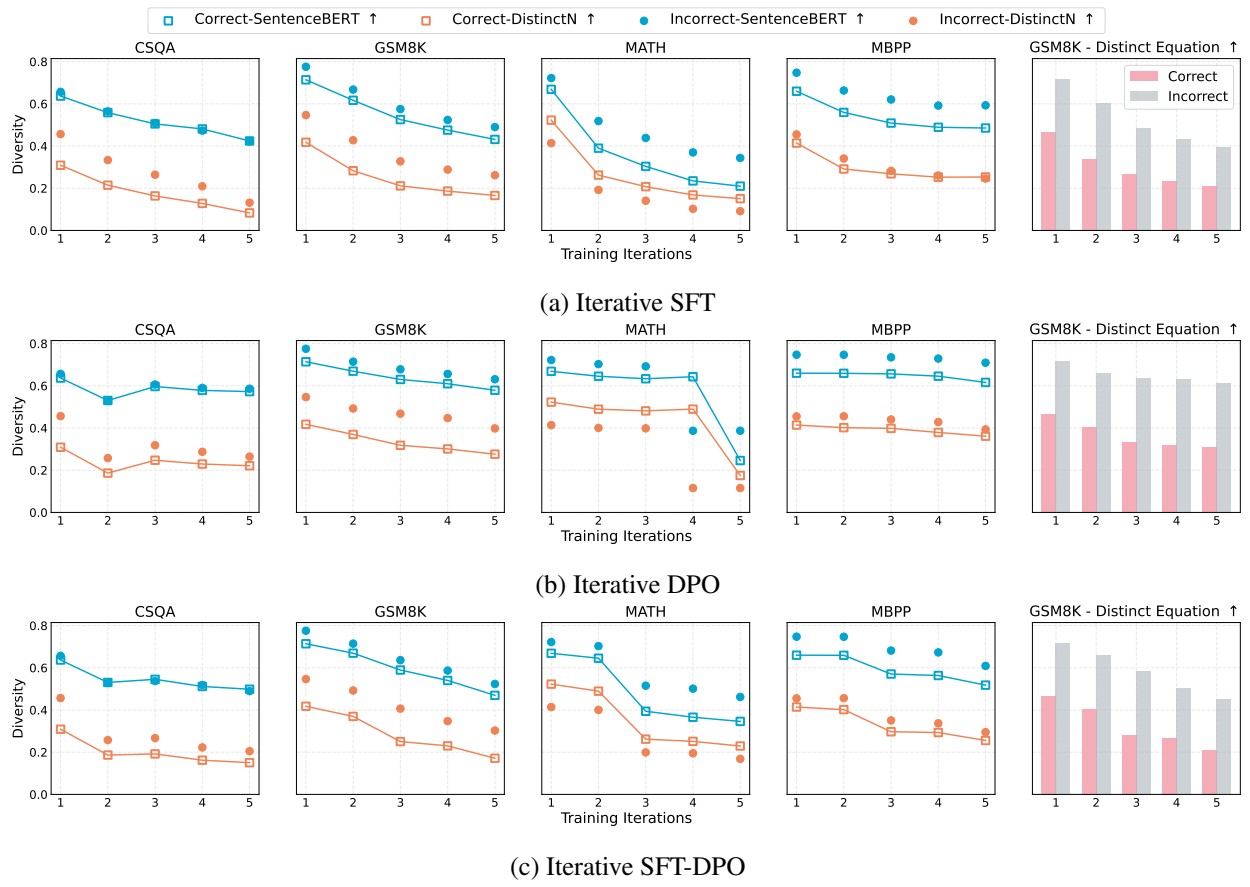

(a) Iterative SFT

(b) Iterative DPO

(c) Iterative SFT-DPO

Figure 4: Diversity of the sampling outputs from $M_t$ within the iterative process.

## 5.1. Improvement Problems

In Figure 1, it is evident when $t > 1$, the pass@1 accuracy of $M_t$ consistently improves in comparison to $M_1$. Traditionally, it has been assumed that this improvement indicates the model progressively learning to tackle more challenging problems (Zelikman et al., 2022). However, we posit a nuanced perspective: while an increase in pass@1 accuracy suggests improvements, it does not inherently equate to an increase in model capabilities to solve more difficult problems.

To better gauge how the model problem-solving capabilities evolve overtime, we propose to first quantify the *improvement problems* as **improvement set** (IS) at each iteration. An intuitive improvement between $\mathcal{M}_t$ and $\mathcal{M}_1$ is the pass@1 accuracy on test set, so we use the subset of test problems that $\mathcal{M}_t$ correctly answers while $\mathcal{M}_1$ fails under greedy decoding to represent IS(t), defined as follows:

$$\text{IS}(t) = \{x \in \mathcal{D}_{\text{test}} \mid M_t(x) = y \wedge M_1(x) \neq y\}. \quad (2)$$

Then we can prompt $M_1$ with the problems in IS(t) and sample $N$ answers for each problem to record the pass@N accuracy of $M_1$. Notably, if $M_1$ exhibits lower pass@N

accuracy even as $N$ increases, it can validate $M_1$ struggles to solve the problems in IS(t) and the iterative process enhances the model's problem-solving abilities.

We apply this evaluative methodology to CSQA, GSM8K, MATH and MBPP datasets with three post-training methods. Generation sampling $N$ varies from $2^1$ to $2^6$ with the temperature set as $0.75$.

**Reversal Observation** As depicted in Figure 3, contrary to prior assumptions, the rapid increase in pass@N accuracy with increasing $N$ challenges the notion of progressively harder problem-solving. Specifically, as $N$ grows, $M_1$ achieves near-perfect pass@N accuracy on IS(t) across all evaluated datasets, suggesting its inherent capacity to tackle the deemed *improvement problems*.

**Selection Optimization for Answer Alignment** The empirical findings depicted in Figure 3 offer a critical insight: iterative self-improvement hardly entails the acquisition of new problem-solving abilities, but rather the enhancement of the model's correct answer selection within its generation space.

## 5.2. Solutions Diversity

While pass@1 accuracy measures the correctness of the final answer, it does not capture the diversity of solutions a model can generate. We posit that a model's capacity to produce diverse solutions is indicative of its robustness and flexibility in problem-solving. To thoroughly understand the evolution of answer diversity during the process of iterative self-improvement, we employ a combination of **Distinct N-grams** (Li et al., 2016) and **Sentence-BERT embedding cosine similarity** (Reimers & Gurevych, 2019) to measure mod diversity. These metrics have been shown to correlate well with human assessments of diversity (Tevet & Berant, 2021). Additionally, for mathematical reasoning, we introduce **Distinct Equations** to measure the diversity of mathematical answers by analyzing the variety of equations in the generated solutions.

Each diversity metric $Div$ takes a set of $N$ model outputs, and produces a scalar score representing how diverse the set is. *Distinct N-grams* measures syntactic diversity by counting the number of unique n-grams (averaged over $n = 1 \ldots 5$) in the output set. The *Sentence-BERT* metric assesses semantic diversity by embedding each output using a sentence transformer and calculating the average cosine similarity between embeddings. The metric is then 1 minus the average similarity, ensuring that higher scores reflect greater diversity. *Distinct Equations*, a specialized metric for mathematical reasoning, computes logical diversity by extracting all equations from the outputs and calculating the proportion of unique equations.

At each iteration, we sample $N = 50$ outputs per problem with a temperature of 0.75. Outputs are categorized into correct and incorrect based on the final answer's correctness. Then for each problem, we use the metric $Div$ to calculate the average diversity for the correct and incorrect answers.

**Reversal Observation** Figure 4 presents the diversity results of three post-training methods during the iterative process. All methods show a consistent decrease in diversity, significantly diminishing the diversity of model outputs over iterations, impacting both correct and incorrect answers. This reduction is evident across all three metrics: syntactic, semantic, and logical diversity. Moreover, comparing Iterative SFT and Iterative DPO, it is clear that both methods exhibit a reduction in diversity, but the extent and pattern of reduction vary. For instance, Iterative DPO maintains a slightly higher semantic diversity (as measured by cosine similarity) over multiple iterations compared to Iterative SFT.

**Trade-Off with Output Diversity.** The evaluation results highlight a critical trade-off in iterative self-improvement: while aiming for higher accuracy, the diversity of outputs, which can be crucial for creativity and robustness

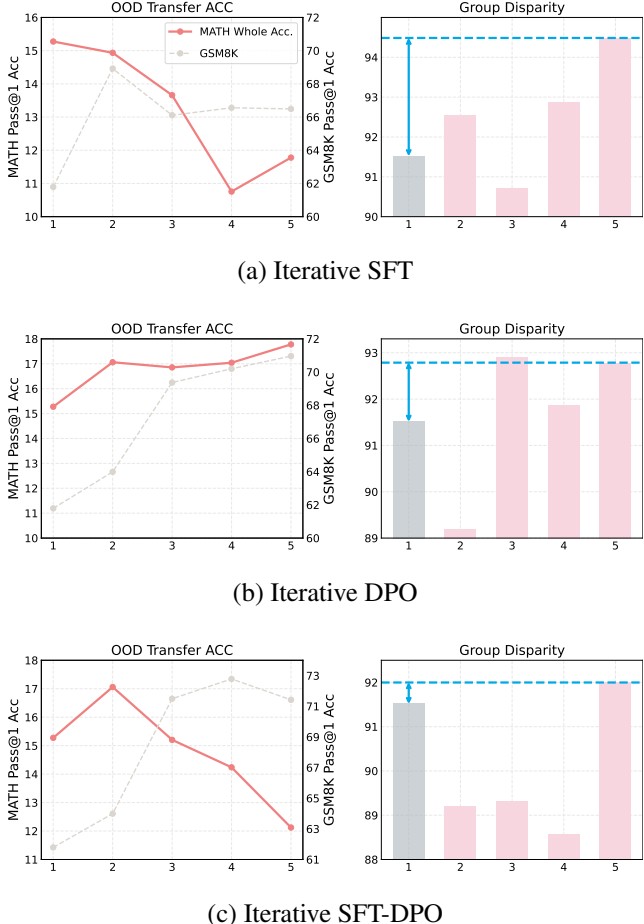

(a) Iterative SFT

(b) Iterative DPO

(c) Iterative SFT-DPO

Figure 5: OOD Performance of $M_t$ on the MATH Algebra Test Set (Post-Training on GSM8K).

in problem-solving, is compromised. Future approaches should consider strategies to maintain or even enhance diversity while improving accuracy.

### 5.3. OOD Generalization

In our pursuit to understand the broader implications of iterative self-improvement, it is crucial to assess not only the models' performance on specific benchmarks but also their ability to generalize to out-of-distribution (OOD) tasks. Generalization performance provides insight into the robustness and adaptability of the models when faced with new and varying types of problems.

To evaluate the generalization capability of the models, we conducted iterative post-training on the GSM8K dataset and then transferred these models to the MATH algebra test set. The MATH algebra test set is organized into five levels of increasing difficulty, providing a comprehensive spectrum to analyze how well the models perform across groups with varied complexities.

For the sake of measuring OOD generalization, we define two metrics as follows defined to facilitate a deeper analysis:

- **Whole Accuracy (Whole Acc.):** This metric represents the pass@1 accuracy across the entire test set, encompassing all difficulty levels from Level 1 to Level 5.

- **Group Disparity:** This metric quantifies the difference in pass@1 accuracy between the best-performing group (Level 1 test set) and the worst-performing group (Level 5 test set), thus highlighting disparities in model performance across different difficulty levels. It is calculated using the following equation:

$$\text{Group Disparity} = \frac{Pass@1(\text{Level 1}) - Pass@1(\text{Level 5})}{Pass@1(\text{Level 1})}$$

A higher value of Group Disparity indicates that the model is performing significantly better on the easier Level 1 while its performance deteriorates on the harder Level 5 group.

**Reversal Observation** As results shown in Figure 5, with the increase in iterative steps, Iterative SFT and Iterative SFT-DPO can significantly harm the OOD generalization. In contrast, Iterative DPO demonstrates a noticeable improvement, which may indicate better generalization to the OOD test set, in consistent with the recent findings that DPO can improve OOD generalization (Kirk et al., 2024). However, our more detailed examination of the results across Group Disparity shows Iterative DPO is widening the performances between the easier and harder groups. This comparison uncovers the OOD performance improvement from Iterative DPO actually stems from fitting simpler problems, iat the expense of solving more complex ones.

**Capabilities Collapse** All three iterative post-training methods can exacerbate the generalization disparities across groups, inadvertently causing models to focus on easier problems rather than enhancing their ability to solve more complex ones. As discussed in Section 5.2, the decrease in solution diversity during iterations may be the bottleneck leading to reduced OOD generalization and capability collapse. This highlights the intricate nature of model capabilities under self-improvement, where capabilities at different levels and different facets will compromise each other. Therefore, research developing more sophisticated methods should employ such a comprehensive, fine-grained evaluative framework to monitor post-training processes, as an increase in a single facet of accuracy does not necessarily represent true self-improvement.

## 6. Conclusion

In this paper, we foster a comprehensive understanding of the current landscape of post-training practices in self-improvement. Our evaluation, beyond simple pass@1 accuracy, utilizing multifaceted metrics such as improvement

problems, solutions diversity and OOD generalization, underscores the necessity for a critical examination of both the progressive and regressive effects in current self-improving post-training methods. By broadening the scope of our analysis, we provide deeper insights into the true nature of iterative self-improvement with post-training, paving the way for more robust and genuinely self-improving LLMs.

## Impact Statement

The current landscape of post-training practices for self-improvement in large language models (LLMs) necessitates a thorough understanding to address both their progressive and regressive effects. In this work, we conduct an in-depth evaluation beyond simple pass@1 accuracy, utilizing multifaceted metrics such as improvement problems, solutions diversity, and OOD generalization. Our findings underscore the critical need for a comprehensive examination of these methods. By expanding our analytical scope, we provide deeper insights into the true nature of iterative self-improvement through post-training, paving the way for more robust and genuinely self-improving LLMs. This research aims to empower scholars and engineers to develop more reliable and effective post-training strategies, ultimately advancing the field of LLMs and pushing the boundaries of model capabilities to tackle more complex challenges.

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

# Appendix

# A. Algorithmic Overview of Post-training

## A.1. Supervised Fine-tuning

Supervised fine-tuning (SFT) is employed to tailor a pre-trained LLM to specific downstream tasks. Consider the training dataset $\mathcal{D} = \{x^{(i)}, y^{(i)}\}_{i=1}^{N}$, where $x^{(i)}$ is the problem and $y^{(i)}$ is the target response, which the model $M$ parameterized by $\theta$ is trained to generate. The training objective of SFT is to minimize the following negative log-likelihood of the answers:

$$\mathcal{L}_{\text{SFT}}(\theta) = -\mathbb{E}_{(x,y) \sim \mathcal{D}} \log p(y|x; \theta) \tag{3}$$

where $p(y|x)$ is the probability of observing the answer $y$ given the problem context $x$.

## A.2. Preference Learning

Preference learning is commonly used to train large language models to learn human preferences. The preference learning dataset includes not only problem and target response pairs but also preferences or rankings between different target responses for the given problem. A typical form of preference learning data is represented as $\mathcal{D} = \{x^{(i)}, y_w^{(i)}, y_l^{(i)}\}_{i=1}^{N}$, where each piece of data contains a problem $x^{(i)}$, and corresponding preferred and dispreferred responses, denoted $y_w^{(i)}$ and $y_l^{(i)}$, respectively. Using a theoretical model of human discrete choice such as the Bradley-Terry model, which relates discrete choices to implicit goodness scores of the underlying options, we can train a reward model with maximum likelihood using this preference data. For the Bradley-Terry model, the reward modeling loss is:

$$\mathcal{L}_R(R_\phi, \mathcal{D}) = -\mathbb{E}_{(x, y_w, y_l) \sim D}[\log \sigma(r_\phi(x, y_w) - r_\phi(x, y_l))]. \tag{4}$$

In the context LLMs, $r_\phi(x, y)$ is initialized from the SFT model $\phi_{\text{SFT}}$. Then, the learned reward function is used to provide feedback to the language model, through the optimization problem described below to train preferences in the language model:

$$\max_{\pi_\theta} \mathbb{E}_{x \sim \mathcal{D}, y \sim \pi_\theta(y|x)}[R_\phi(x, y)] - \beta \mathbb{D}_{KL}[\pi_\theta(y|x)||\pi_{\text{SFT}}(y|x)], \tag{5}$$

where $\beta$ is a parameter controlling the deviation from the base reference policy $\pi_{\text{SFT}}$. More recently, (Rafailov et al., 2023) show that the optimal policy for the learned reward can be extracted in closed form, especially skipping the need to perform iterative, approximate policy learning. The resulting algorithm, direct preference optimization (DPO), is simpler to tune and less computationally demanding than prior methods, while optimizing the same objective. We therefore use DPO as the algorithm for the implementation for preference learning in our experiments. The DPO loss for the language model policy $\pi_\theta$ is

$$\mathcal{L}_{\text{DPO}} = -\mathbb{E}_{(x, y_w, y_l) \sim D_p} \left[ \log \left( \sigma \left( \beta \log \frac{\pi_\theta(y_w \mid x)}{\pi_{\text{SFT}}(y_w \mid x)} - \beta \log \frac{\pi_\theta(y_l \mid x)}{\pi_{\text{SFT}}(y_l \mid x)} \right) \right) \right]. \tag{6}$$

# B. Experiments

## B.1. Training Details

We use a fully fine-tuning setting for training LLaMA2-7B, Mistral-7B and LLaMA3-8B models either for supervised and preference fine-tuning. All training experiments are conducted on 8 NVIDIA A100 GPUs, and all experiments collectively consumed approximately 2000 A100 GPU hours. Our training codebase is based on LLaMA Factory (Zheng et al., 2024), and we use vLLM (Kwon et al., 2023) framework to perform inference for both CoT sampling and test evaluation. Detailed hyperparameters utilized throughout these experiments are documented in Table 1.

## B.2. Dataset Details

**CommonsenseQA** (Talmor et al., 2019) offers a collection of 5-way multiple-choice questions on commonsense knowledge scenarios. It contains 12,102 questions with training/validation/testing set splits. Due to the unavailability of correct answers for the testing set, we utilize the validation set comprising 1,221 questions for evaluation, following the practice of (Kojima et al., 2022).

**GSM8K** (Cobbe et al., 2021) consists of 8.5K high-quality grade school math problems created by human problem writers, with the segmentation into 7.5K training problems and 1K test problems. These problems take between 2 and 8 steps to

| Type | Parameter | Value |
|------|-----------|-------|
| Supervised Fine-Tuning | Batch Size | 128 |
| | Learning Rate {LLaMA2-7B} | $1e-5$ |
| | Learning Rate {Mistral-7B, LLaMA3-8B} | $2e-6$ |
| | Learning Rate Scheduler | Cosine |
| | Warm-up Ratio | 0.03 |
| | Optimizer | AdamW |
| | Epoch | 3 |
| Preference Fine-Tuning | Batch Size | 32 |
| | Learning Rate {LLaMA2-7B} | $2e-6$ |
| | Learning Rate {Mistral-7B, LLaMA3-8B} | $2e-7$ |
| | KL Coefficient ($\beta$) | 0.3 |
| | Optimizer | AdamW |
| | Epoch | 1 |
| Sampling Generation | Temperature | 0.75 |
| | Top_$p$ | 0.95 |
| | Top_$k$ | 50 |
| | Max_tokens | 512 |
| Evaluation Generation | Temperature | 0 |
| | Top_$k$ | -1 |
| | Max_tokens | 512 |

Table 1: Hyperparameters in all the experiments.

solve, and solutions primarily involve performing a sequence of elementary calculations using basic arithmetic operations (+ - / *) to reach the final answer.

**MATH** (Hendrycks et al., 2021) offers high school math competition problems that span seven subjects including Prealgebra, Algebra, Number Theory, Counting and Probability, Goemetry, Intermediate Algebra and Precalculus. It consists of 7,500 and 5,000 samples for training and testing, respectively. Compared to GSM8K, addressing MATH challenges involves more intricate and extended steps.

**MBPP** (Austin et al., 2021) consists of around 1,000 crowd-sourced Python programming problems, designed to be solvable by entry-level programmers, covering programming fundamentals, standard library functionality, and so on. Each problem consists of a task description, code solution and 3 automated test cases. Following the experimental setup described in (Austin et al., 2021), we utilize Task IDs 11-510, comprising 500 problems, as our test set. The remaining 374 problems, ranging from Task IDs 601 to 974, are employed for fine-tuning purposes.

### B.3. Evaluation Protocols

**Zero-shot Prompting** We employ zero-shot prompts, as listed in Figure 6 for answer sampling and evaluation tests. Our comprehensive evaluation across all benchmarks demonstrates that zero-shot prompting not only reduces inference costs but also consistently outperforms few-shot prompting in terms of performance. Consequently, when LLMs are fine-tuned for task-specific applications, we advocate for the adoption of zero-shot prompting as a superior method compared to various few-shot techniques. This perspective aligns with the findings of (Yu et al., 2024), who also reported the advantages of zero-shot over few-shot prompting for fine-tuned LLMs.

**Generation Diversity** In evaluating natural language generation (NLG) models, two prevalent methods for assessing output diversity are the n-gram-based metric and the embedding-based metric, which embeds generated sentences in a latent space. In this paper, we adopt distinct n-grams (Li et al., 2016) and Sentence-BERT Embedding Cosine Similarity (Reimers & Gurevych, 2019) metrics.

**Distinct n-grams** is a straightforward yet effective method to quantify the lexical diversity of generated text. This metric calculates the proportion of unique n-grams (sequences of n words) within the generated text. The distinct n-gram measure is typically computed for unigrams, bigrams, trigrams, and sometimes higher-order n-grams. Mathematically, for a generated

Figure 6: Zero-shot Evaluation Prompt.

sequence $S$, distinct-n is defined as:

$$\text{distinct-n}(S) = \frac{|\text{unique n-grams in } S|}{|\text{total n-grams in } S|} \tag{7}$$

This measure provides a direct indication of how varied the vocabulary and phrases are within the generated text. In general , higher distinct-n values indicate greater diversity.

**Sentence-BERT** embedding cosine similarity assesses the semantic diversity of generated sentences. Sentences generated by the model are first encoded into embeddings using Sentence-BERT. The cosine similarity between each pair of sentence embeddings is then computed. Cosine similarity between two embeddings $\mathbf{u}$ and $\mathbf{v}$ is given by:

$$\cos(\mathbf{u}, \mathbf{v}) = \frac{\mathbf{u} \cdot \mathbf{v}}{\|\mathbf{u}\|\|\mathbf{v}\|} \tag{8}$$

The average cosine similarity of all sentence pairs gauges the overall semantic similarity. Lower average cosine similarity indicates higher semantic diversity, as the sentences are less similar in meaning. In our calculations, we measure diversity using $1 -$ average cosine similarity, ensuring that higher values reflect greater semantic diversity.

**Distinct Equations** provides a direct indication of how varied the mathematical approaches and solutions are within the generated text. The calculation of this metric involves two steps that identifies all equations present in the generated mathematical reasoning steps first, and then computes the ratio of unique equations to the total number of equations. Mathematically, for a set of generated equations $E$, distinct equations is defined as:

$$D_{eq}(E) = \frac{|\text{unique equations in } E|}{|\text{total equations in } E|} \tag{9}$$

Higher $D_{eq}$ values indicate greater diversity in the mathematical reasoning processes.

## C. Extrapolation Analysis

In Section 4.2, we mentioned that *correct answer coverage* may be a deeper factor influencing the subsequent improvements in iterative self-improvement. Here, we provide a detailed explanation of the related concepts involved in this influencing factor, as well as the derivation of the calculation for correct answer coverage (in Equation 1) as presented in this paper.

Additionally, we emphasize that our consideration of coverage as a deeper factor is a preliminary conclusion drawn from summarizing the factors of the model ($M$), post-training function ($\mathcal{F}$), and task dataset ($\mathcal{D}$) and the empirical results validated in Figure **??**. It should be noted that we need further work and more extensive experiments to both theoretically and

empirically validate this observation, as the discussion of correct answer coverage is beyond our work. Our intention here is to offer empirical insights and lay the groundwork for future investigations into this aspect of iterative self-improvement.

**Answer space**: For a given dataset $\mathcal{D}$ (test set), all possible (query, answer) pairs form the answer space of the test set. Here, the set of query is fixed, and for a given query, the number of possible answers can be quite large, hence we call the space as the *answer space*. Naturally, the entire answer space can be partitioned into a correct answer space and an incorrect answer space based on whether the answers are correct. In practical experiments, the correctness of an answer is approximated by whether its final result exactly matches the final result provided by the ground truth in the training set.

**Answer distribution**: For a given model $M$, its answer distribution refers to the probability distribution of generating answers conditioned on queries from dataset $D$. For a specific element $(q_i, a_{ij}), q_i \in \mathcal{D}_{\text{test}}$ in the answer space, the generation of this (query, answer) pair occurs in two steps: first, sampling $q_i$ from all queries in $D$, then model M generates $a_{ij}$ conditioned on $q_i$. Therefore, the probability at $(q_i, a_{ij})$ is the product of the probability of sampling $q_i$ from all queries in $D$ and the probability of model $M$ generating $a_{ij}$ conditioned on $q_i$. Considering that all queries should have equal importance, we define that all queries are sampled with the same probability, which is $\frac{1}{\mathcal{D}_{\text{test}}}$. The answer distribution of $M$ can be mathematically linked to model $M$ as follows:

$$P_M(q_j, a_{ij}) \overset{\text{def}}{=} M(a_{ij}|q_j)P(q_j) = M(a_{ij}|q_j)\frac{1}{\mathcal{D}_{\text{test}}} \tag{10}$$

which $P_M$ represents the model's answer distribution, and $M(a_{ij}|q_j)$ denotes the probability of model $M$ generating $a_{ij}$ conditioned on $q_j$.

**Correct Answer Coverage**: As mentioned earlier, the Correct Answer Coverage represents the correctness rate of all answers generated by model $M$ on a dataset $D$ (training set). It can be calculated using the following mathematical formula:

$$\text{Correct Answer Coverage} = \int_{\text{Correct Answer Space}} P_M(a, q) \tag{11}$$

Although we cannot exhaust the entire answer space and calculate a probability distribution to demonstrate the trend of the answer distribution in the progress of self-improvement, we can get an unbiased estimate of it by sampling answers and calculate the ratio of the number of correct answers to the total number of answers generated by model $M$ for all queries in $\mathcal{D}_{\text{test}}$, where $N$ answers are generated for each query, as illustrated in equation 1. The proof of Equation 1 is as follows:

$$
\begin{aligned}
\text{Correct Answer Coverage} &= \int_C P_M(a, q) \\
&= \int_C \frac{1}{|\mathcal{D}_{\text{test}}|} P_M(a|q) \\
&= \frac{1}{|\mathcal{D}_{\text{test}}|} \int_C P_M(a|q) \\
&= \frac{1}{|\mathcal{D}_{\text{test}}|} \int_C \sum_{x \in \mathcal{D}_{\text{test}}} P_M(a|q = x) \\
&= \frac{1}{|\mathcal{D}_{\text{test}}|} \sum_{x \in \mathcal{D}_{\text{test}}} \int_C P_M(a|q = x) \\
&= \frac{1}{|\mathcal{D}_{\text{test}}|} \sum_{x \in \mathcal{D}_{\text{test}}} \mathbb{E}(\frac{N_x^c}{N}) \\
&= \mathbb{E}(\frac{1}{|\mathcal{D}_{\text{test}}|} \sum_{x \in \mathcal{D}_{\text{test}}} \frac{N_x^c}{N}) \\
&= \mathbb{E}[\frac{1}{|\mathcal{D}_{\text{test}}|} \sum_{x \in \mathcal{D}_{\text{test}}} \frac{1}{N} \sum_{i=1}^N \mathbb{I}[M(x_i) == y]],
\end{aligned}
\tag{12}
$$

where $C$ denotes the correct answer space, and $N_x^c$ represents the number of correct answers generated by model conditioned on the given query $x$.

## D. Case Study

In this section, we select one problem from GSM8K test set and record the outputs of Mistral-7b during the process of iterative DPO. This real case can vividly display our findings from Section 5, which suggest that iterative self-improvement primarily manifests as an optimization of answer selection rather than substantial enhancements in problem-solving capabilities. Additionally, we will clearly see how diversity of the model's reasoning steps decreases throughout the process.

**Problem**: According to its nutritional info, a bag of chips has 250 calories per serving. If a 300g bag has 5 servings, how many grams can you eat if your daily calorie target is 2000 and you have already consumed 1800 calories?

**Label**: If the total calorie target is 2000 and I have consumed 1800 calories then I have 2000-1800 = <<2000-1800=200>>200 calories left to eat\nIf each serving of chips has 250 calories and I only have 200 calories left to eat, then I can only eat 200/250 of a serving = 4/5 of a serving\nWe also know that a 300g bag of chips has 5 servings, hence each serving has 300g/5 = <<300/5=60>>60 grams\nIf I can only eat 4/5 of a serving, then I can eat only 60g * 4/5 = 240g/5 = 48 grams\n#### 48

**Iteration 1**

**Greedy Decoding Answer**

You have 2000-1800 = <<2000-1800=200>>200 calories left for the day.\nSince each serving has 250 calories, 5 servings have 250*5 = <<250*5=1250>>1250 calories.\nThus, you can eat 1250-200 = <<1250-200=1050>>1050 grams of chips.\n#### 1050 ✖

**Sampling Answers**

A bag of chips has 5 servings, and you can eat 700/250 = <<700/250=2.8>>2.8 servings.\nSince a serving is 150g, you can eat 2.8*150 = <<2.8*150=420>>420g of chips.\n#### 420 ✖

A 300g bag has 5 servings and each serving contains 250 calories so 250*5 = <<5*250=1250>>1250 calories\nIf your daily calorie target is 2000 and you have already consumed 1800 calories then you still have 2000-1800 = <<2000-1800=200>>200 calories left for the day.\nA 300g bag has 1250 calories and you still have 200 calories left so you can eat 1250-200 = <<1250-200=1050>>1050g\n#### 1050 ✖

If a serving has 250 calories, then the entire bag has 250*5 = <<250*5=1250>>1250 calories.\nYour daily calorie target is 2000, and you have already consumed 1800, so you still have 2000-1800 = <<2000-1800=200>>200 calories.\nSince there are 1250 calories in the bag of chips, then you can eat 200/1250 = <<200/1250=0.16>>0.16 of the bag.\nThere are 300g in the bag, so 0.16 of the bag is 300*0.16 = <<300*0.16=48>>48g\n#### 48 ☑

A serving has 250 calories so 5 servings have 5*250 = <<250*5=1250>>1250 calories\nYou have already consumed 1800 calories so you have 2000-1800 = <<2000-1800=200>>200 calories left to consume
You can eat 200/250 = <<200/250=0.8>>0.8 servings\nOne serving is 150g so you can eat 0.8*150 = <<0.8*150=120>>120 grams\n#### 120 ✖

2000-1800 = <<2000-1800=200>>200\n200/250 = <<200/250=0.8>>0.8\nYou can eat 0.8*300 = <<0.8*300=240>>240 grams of chips.\n#### 240 ✖

**Iteration 2**

**Greedy Decoding Answer**

A bag of chips has 250 calories per serving and there are 5 servings in a bag, so a bag has 250*5 = <<250*5=1250>>1250 calories\nYou have already consumed 1800 calories and your daily calorie target is 2000, so you have 2000-1800 = <<2000-1800=200>>200 calories left\nA bag of chips has 1250 calories and you have 200 calories left, so you can eat 1250-200 = <<1250-200=1050>>1050 grams of chips\n#### 1050 ✖

**Sampling Answers**

If a serving has 250 calories, then a bag of chips has 250*5=<<250*5=1250>>1250 calories\nIf you have already consumed 1800 calories, then you have 2000-1800=<<2000-1800=200>>200 calories left for the day.\nTherefore, you can eat 200/250=<<200/250=0.8>>0.8 of a bag of chips.\nIn grams, this would be 0.8*300=<<0.8*300=240>>240\n#### 240 ✖

A serving has 250 calories and a bag has 5 servings, so the bag has 250*5 = <<250*5=1250>>1250 calories\nYour daily calorie target is 2000 and you have already consumed 1800, so you can still eat 2000-1800 = <<2000-1800=200>>200 calories\nA bag of chips has 1250 calories and you can still eat 200 calories, so you can eat 1250-200 = <<1250-200=1050>>1050g of chips\n#### 1050 ✖

Since a bag of chips has 5 servings and each serving has 250 calories, the whole bag has 5*250 = <<5*250=1250>>1250 calories\nIf you have already consumed 1800 calories and your daily calorie target is 2000, then you can still eat 2000-1800 = <<2000-1800=200>>200 calories\nSince a bag of chips has 1250 calories, then you can eat 200/1250 = <<200/1250=0.16>>0.16 of a bag\nSince a bag of chips weighs 300g and you can only eat 0.16 of a bag, then you can eat 0.16*300 = <<0.16*300=48>>48g\n#### 48 ☑

A serving has 250 calories so 5 servings have 5*250 = <<250*5=1250>>1250 calories\nYou have already consumed 1800 calories so you have 2000-1800 = <<2000-1800=200>>200 calories left to consume
You can eat 200/250 = <<200/250=0.8>>0.8 servings\nOne serving is 150g so you can eat 0.8*150 = <<0.8*150=120>>120 grams\n#### 120 ✖

**Iteration 3**

**Greedy Decoding Answer**

If a serving has 250 calories, 5 servings will have 250*5 = <<250*5=1250>>1250 calories\nIf you have already consumed 1800 calories, you can still eat 2000-1800 = <<2000-1800=200>>200 calories\nSince 1 serving has 250 calories, you can eat 200/250 = <<200/250=0.8>>0.8 servings\nSince each serving is 50g, you can eat 0.8*50 = <<0.8*50=40>>40g\n#### 40 ✗

**Sampling Answers**

If a serving has 250 calories, then a bag of chips has 250*5=<<250*5=1250>>1250 calories\nIf you have already consumed 1800 calories, then you have 2000-1800=<<2000-1800=200>>200 calories left for the day.\nTherefore, you can eat 200/250=<<200/250=0.8>>0.8 of a bag of chips.\nIn grams, this would be 0.8*300=<<0.8*300=240>>240\n#### 240 ✗

If a 300g bag has 5 servings, each serving is 300/5 = <<300/5=60>>60g\nThe nutritional info says a serving has 250 calories, so 60g has 250*60/60 = <<250*60/60=250>>250 calories\nIf you have already consumed 1800 calories, you have 2000-1800 = <<2000-1800=200>>200 calories left.\nSince 60g has 250 calories, you can eat 200/250 = <<200/250=0.8>>0.8*60 = <<0.8*60=48>>48g of chips.\n#### 48 ✓

If a serving has 250 calories, then the whole bag has 250*5 = <<250*5=1250>>1250 calories.\nIf you have already consumed 1800 calories, then you can eat 2000-1800 = <<2000-1800=200>>200 calories worth of chips.\nSince there are 1250 calories in the whole bag, then 200 calories is equal to 200/1250 = 1/6 of the bag.\nTherefore, you can eat 300/6 = <<300/6=50>>50 grams of chips.\n#### 50 ✗

---

**Iteration 4**

**Greedy Decoding Answer**

If a bag of chips has 250 calories per serving, and a 300g bag has 5 servings, then the 300g bag has 250*5 = <<250*5=1250>>1250 calories.\nIf you have already consumed 1800 calories, then you can eat 2000-1800 = <<2000-1800=200>>200 calories worth of chips.\nSince each chip has 1250 calories, you can eat 200/1250 = <<200/1250=0.16>>0.16 bags of chips.\nSince each bag of chips has 300g, you can eat 0.16*300 = <<0.16*300=48>>48 grams of chips.\n#### 48 ✓

**Sampling Answers**

If a serving has 250 calories, then a bag of chips has 250*5=<<250*5=1250>>1250 calories\nIf you have already consumed 1800 calories, then you have 2000-1800=<<2000-1800=200>>200 calories left for the day.\nTherefore, you can eat 200/250=<<200/250=0.8>>0.8 of a bag of chips.\nIn grams, this would be 0.8*300=<<0.8*300=240>>240\n#### 240 ✗

If a 300g bag has 5 servings, each serving is 300/5 = <<300/5=60>>60g.\nIf a bag of chips has 250 calories per serving, then 60g of chips have 250*60/60=250>>250 calories.\nIf you have already consumed 1800 calories, then you can eat 2000-1800 = <<2000-1800=200>>200 calories worth of chips.\nSince 60g of chips have 250 calories, then 200/250 = <<200/250=0.8>>0.8 servings of chips can be eaten.\nTherefore, you can eat 0.8*60 = <<0.8*60=48>>48g of chips.\n#### 48 ✓

A serving has 250 calories and there are 5 servings in the bag, so the whole bag has 250*5 = <<250*5=1250>>1250 calories.\nYou have already consumed 1800 calories and your daily target is 2000, so you can still consume 2000-1800 = <<2000-1800=200>>200 calories.\nThe bag has 1250 calories and you can still consume 200 calories, so you can eat 1250-200 = <<1250-200=1050>>1050 grams.\n#### 1050 ✗

---

**Iteration 5**

**Greedy Decoding Answer**

If a serving has 250 calories, and a bag has 5 servings, then the bag has 250*5 = <<250*5=1250>>1250 calories.\nIf you've already consumed 1800 calories, then you can eat 2000-1800 = <<2000-1800=200>>200 calories worth of chips.\nSince each chip has 1250 calories, then you can eat 200/1250 = <<200/1250=0.16>>0.16 bags of chips.\nSince a bag has 300g, then you can eat 0.16*300 = <<0.16*300=48>>48g of chips.\n#### 48 ✓

**Sampling Answers**

If a 300g bag has 5 servings, each serving is 300/5 = <<300/5=60>>60g.\nIf your daily calorie target is 2000 and you have already consumed 1800 calories, you can consume 2000-1800 = <<2000-1800=200>>200 calories more.\nSince each serving has 250 calories, you can consume 200/250 = <<200/250=0.8>>0.8 servings more.\nSince each serving is 60g, you can consume 0.8*60 = <<0.8*60=48>>48g more.\n#### 48 ✓

If a 300g bag has 5 servings, each serving is 300/5 = <<300/5=60>>60g.\nIf your daily calorie target is 2000 and you have already consumed 1800 calories, you have 2000-1800 = <<2000-1800=200>>200 calories left.\nSince a bag of chips has 250 calories per serving, you can eat 200/250 = <<200/250=0.8>>0.8 servings.\nSince each serving is 60g, you can eat 0.8*60 = <<0.8*60=48>>48g of chips.\n#### 48 ✓

A bag of chips has 250 calories per serving, and there are 5 servings in a 300g bag, so the 300g bag has 250*5 = <<250*5=1250>>1250 calories.\nIf a 300g bag has 1250 calories, and you have already consumed 1800 calories, then you can eat 1250-1800 = <<1250-1800=-550>>-550 calories worth of chips.\nSince there are 1000 calories in 1 kilogram, then -550 calories is equal to 1000*-0.55 = <<1000*-0.55=550>>550 grams.\n#### 550 ✗

Figure 7: One case of sampled responses from the test set after iterative DPO training of the Mistral-7B model on the GSM8K dataset.

