# OpenReview forum: "Progress or Regress? Self-Improvement Reversal in Post-training"
_ICML.cc/2024/Workshop/AI4MATH — ICML 2024 Workshop AI4MATH Oral_

### Official Review · Reviewer_qCiQ · 2024-06-10

**Rating:** 7
**Confidence:** 4

**Summary:**

This paper evaluates a sub-family of self-improvement methods in the field of LLM for math reasoning. It proposes a new systematic multi-facet way to evaluate the model's reasoning capability improvements, and challenges the beliefs of existing works that better PASS@1 accuracy equals to better reasoning capability of the model.

**Questions:**

Typos: line 368 "mod diversity"; line 408 "iat"; line 611 incorrect latex format; line 741 starting with a comma.

**Reasons To Accept:**

The question raised by the paper is critical and insightful.

The paper is logically sound and coherent. It provides strong theoretical and experimental reasons to embrace a comprehensive set of metrics to evaluate mathematical reasoning capabilities. The paper is also very well structured.

**Reasons To Reject:**

The authors conducted a limited literature review. Although the paper's title sounds like a common problem, both the literature review and the experimental setups essentially only focus on iterative self-improvement methods using SFT or DPO, which does not account for all variants of self-improvement methods.

Some references consider adding:\
[1] Learning Math Reasoning from Self-Sampled Correct and Partially-Correct Solutions, ICLR 2023.

Their method "can greatly improve PASS@k, especially for higher k values". However, their approach "does not improve PASS@1, which is expected as learning from multiple targets mainly helps with increasing the diversity of the sampled solutions rather than improving the most-probable solution". This paper shows that self-improvement can indeed **improve diversity** if the self-improvement data includes partially correct solutions, but not simultaneously improving accuracy.

[2] Query and Response Augmentation Cannot Help Out-of-domain Math Reasoning Generalization.

Their paper aligns with the narrative of this paper, saying that even learning from GPT4 **cannot** improve generalization from GSM8K to MATH, let alone learning from the model itself.

Incorporating these related works somewhere in the paper is not difficult. The authors should provide reasons for why they chose to not include these strongly related works in the paper.

---

### Official Review · Reviewer_LKhf · 2024-06-11

**Rating:** 8
**Confidence:** 4

**Summary:**

Post-training methods, such as iterative preference learning, have received recognition for their ability to enhance the problem-solving capabilities of Large Language Models (LLMs), such as mathematical reasoning, without human intervention. However, upon reviewing the findings, it was discovered that post-training can lead to a phenomenon called self-improvement reversal. Rigorous experimentation and analysis across various problem-solving tasks have revealed empirical results indicating that models demonstrating improved performance on benchmarks may paradoxically experience declines in broader essential capabilities, such as output diversity and out-of-distribution generalization.

**Questions:**

The largest model in the paper is 8B. Is the conclusion the same for lager model, such as 70B?

**Reasons To Accept:**

**Originality:** It is new to know that the effect of post-training: Through rigorous experimentation and analysis
across diverse problem-solving tasks, the empirical results point out the phenomenon of self improvement reversal, where models showing
improved performance across benchmarks will paradoxically exhibit declines in broader, essential capabilities, like output diversity and out-of-distribution generalization.

**Quality:** The paper is well-written with extensive experiments.

**Clarity:** The paper is well-written.

**Significance:** The conclusion and insight about post-training are important.

**Reasons To Reject:**

Figure 1 does not have legend so that it takes more time for reader to know the meaning.

---

### Meta-Review · Area_Chair_dpU5 · 2024-06-13

**Recommendation:** Accept (Oral, top 2)
**Confidence:** 4

**Metareview:**

This paper studies self-improvement through post-training methods for LLM problem-solving.
They propose an evaluation framework to distinguish between superficial metric improvements and genuine enhancement. They conduct experimentation and analysis and find that models showing improved performance across benchmarks will paradoxically exhibit declines in broader and essential capabilities.
The reviewers agree that this research question is critical and insightful (Reviewer qCiQ), and the findings provide new and significant knowledge to the community (Reviewer LKhf). This paper is sound as it includes extensive experiments. And it is well written.

---

### Decision · Program_Chairs · 2024-06-13

Accept (Oral)